# The effects of secretome of umbilical cord mesenchymal stem cells on regeneration of sciatic nerve defects in Sprague dawley rats

Aryadi Kurniawan[1], Ismail Hadisoebroto Dilogo[1,2], Jeanne Adiwinata Pawitan[3], Wahyu Widodo[1], Ihsan Oesman[1], Ade Martinus[1] *

1 Faculty of Medicine Universitas Indonesia, Department of Orthopaedics & Traumatology, Dr Cipto Mangunkusumo National Central Hospital, Jakarta, Indonesia, 2 Stem Cell Medical Technology Integrated Service Unit, Dr Cipto Mangunkusumo National Central Hospital, Jakarta, Indonesia, 3 Faculty of Medicine Universitas Indonesia, Department of Histology, Dr Cipto Mangunkusumo National Central Hospital, Jakarta, Indonesia

* orthoademartinus@gmail.com

## Abstract

### Background and purpose

Current treatments for peripheral nerve defects are suboptimal. Mesenchymal stem cell (MSC) implantation holds promise, with studies indicating their efficacy through the secretome. This study aims to assess the secretome's potency in regenerating peripheral nerve defects.

### Animals and methods

Twenty-eight spraque dawley (SD) rats were divided into four groups, with a 10mm segmental sciatic nerve defect created. Group I received autografts, while Groups II, III, and IV had proximal and distal nerve stumps fixed with a conduit. Group II received MSC implantation, Group III received umbilical cord MSC secretome, and Group IV was treated with silicone conduits. Motoric recovery was assessed using the Sciatic Functional Index (SFI) at 6, 9, and 12 weeks. After 12 weeks, rats were sacrificed for measurements of gastrocnemius muscle weight ratio and sciatic nerve histomorphometry.

### Result

The highest ratio of the gastrocnemius muscle were found in groups 1 and 3, namely 0.7 ± 0.06 and 0.7 ± 0.2 (p <0.001). The highest number of myelinated axons was found in group 1 and 3, namely 175.1 ± 15.9 and 168.9 ± 11.7 (p <0.001). The secretome had the similar effectiveness with autograft in promoting regeneration of the sciatic nerve defect.

### Interpretation

The secretome of the umbilical cord MSC can be a novel therapy in replacing autografts for the management of peripheral nerve defects.

**Data Availability Statement:** Data are available at https://github.com/ademartinus/Umbilical-Cord-

Mesenchymal-Stem-Cell-s-Secretome-on-Sciatic-Nerve-Regeneration.git.

**Funding:** The author(s) received no specific funding for this work.

**Competing interests:** The authors have declared that no competing interests exist.

## Introduction

Severe trauma causing damage to peripheral nervous systems often results in limitations in daily activities. Managing such injuries remains challenging in orthopedics, with current approaches deemed unsatisfactory for improving sensory and motor functions. This poses a significant obstacle to enhancing the quality of life for individuals affected by peripheral nerve injuries [1,2].

When a nerve gap in peripheral nerve injury prevents tension-free primary repair, the gold standard treatment is the nerve autograft procedure. However, this approach has drawbacks, including donor morbidity, donor-recipient incompatibility, limited donor availability, and the potential for painful neuromas. Alternatives like conduits or allografts often fall short, necessitating a new and innovative approach for effective management of peripheral nerve defects [3].

Mesenchymal stem cell (MSC) transplantation has shown promise in promoting neural regeneration in preclinical animal research. However, recent studies indicate low cellular viability when these stem cells are transplanted into the nervous system [3,4]. Furthermore, a significant portion of stem cells failed to differentiate and had a survival period of less than two weeks. This supports the notion that the regenerative benefits of nerve cells may stem from the secretory products of mesenchymal stem cells, comprising growth factors and cytokines collectively known as secretome. This secretome can be extracted from the conditioned medium of mesenchymal stem cells [3,5].

The secretome contains trophic factors such as *nerve growth factor* (NGF), *neurotrophin-3* (NT-3), and *hepatocyte growth factor* (HGF) which are thought to induce the migration of schwann cells, vasculogenesis, regeneration of axons, and protect neurons from apoptosis. Therefore, the secretome can be a novel potential therapy for increasing the regeneration of peripheral nerve defect by providing an optimal environment [6–8].

This study aimed to assess the regenerative potential of secretome in rat models with peripheral nerve defects. The evaluation included motoric function recovery, target muscle weight, and histomorphometric analysis.

## Materials and methods

This was an experimental animal study involving 12-16-week-old male Sprague Dawley rats weighing 250–300 grams. The 28 Sprague-Dawley rats were randomly assigned into 4 different groups. All of them underwent intraperitoneal anesthesia injection by using ketamine and xylazine with doses of 10 mg/kg and 15/mgkg, respectively. Afterward, we performed procedure to make segmental cutting of the right sciatic nerve resulting in 10 mm defect (Fig 1). Group I was performed segmental cut which were stitched afterward as an autograft while in the other groups, proximal and distal nerve ends were fixed with a conduit. Additionally, group II received implantation of one million umbilical cord mesenchymal stem cells (UC-MSC) and group III reveived implantation of UC-MSC secretome; while group IV underwent conduit fixation only. Evaluation of motoric function recovery was assessed by using Sciatic Functional Index (SFI) on week 6 (SFI 1), week 9 (SFI 2), and week 12 (SFI 3). Postoperatively, the rats were given with oral amoxicillin and paracetamol with doses of 100 mg/kg for both drugs. On week 12, the rats were sacrificed by exsanguination. The involved sciatic nerve, left and right gastrocnemius muscles were harvested. Histomorphometry evaluation of harvested sciatic nerve was performed in order to get information about the number and the diameter of the myelinated axon that regenerating in each group. The right and left gastrocnemius muscle wet weight was compared to get the ratio of the target muscle in each group.

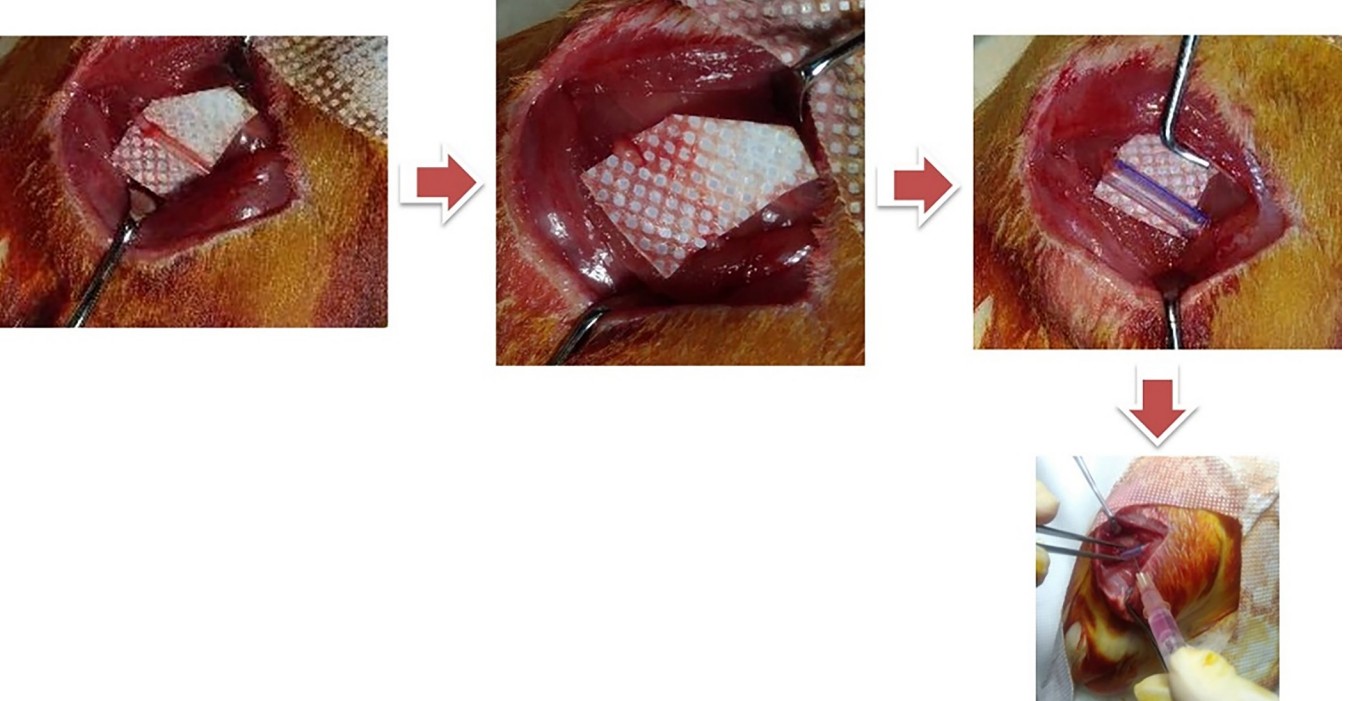

**Fig 1. Surgery procedures of Spraque dawley rats (exposed sciatic nerves, transected to leave a 10 mm nerve defect, and silicon conduit placement).**

Before the surgery, UC-MSC was taken from a human donor. The UC was chopped into size of 2–5 mm then placed in a complete medium consisting of α-MEM (10% human AB serum and 10% autologous / allogenic umbilical cord blood serum), penicillin / streptomycin (100 U / mL), and 10% TC (Indonesian Red Cross). Culture was observed daily using an inverted microscope. The medium was replaced every 3 days. Explants that had grown and 80% confluent then was harvested (usually 5th day). The Secretome was prepared from Conditioned Medium (CM). UC-CM was extracted when the UC-MSC harvested, then it was centrifuged for 30 minutes, in speed of 3500 rpm (revolutions per minute) and precipitate was removed. CM that had been filtered then concentrated and stored at -200˚C. The CM was thawed before the implantation procedure.

The research has been granted ethical approval from the Ethical Committee of the Faculty of Medicine Universitas Indonesia with the admission number of 705/UN2.F1/ETIK/2018.

## Motoric function recovery on walking track analysis

The functional recovery of sciatic nerve was measured using the walking track analysis on week 6, 9, and 12 after the surgery. It was performed on plastic tray with the dimension of 80 x 40 x 25 cm3, sawdust as the base of the animal house, acrylic-based tunnel, red and green ink, and millimeter block paper.

Both feet of SD rats were colored with the red and green ink. The SD rats then walked in the tunnel and the footsteps were blotted onto a millimeter block paper that covered the base of the tunnel. The result on the millimeter block paper was measured in order to obtain the SFI (Sciatic Functional Index). SFI is a quantitative method with numerical scale, which is used to assess the level of the sciatic nerve defect recovery on the experimental rats. The measured components were the distance between the heel and the tip of the third toe (PL, Print

Length), the distance between the first toe and the fifth toe (TS, Toe Spread), and the distance between the second and the fourth toe (IT, Intermediary Toe Spread). The measurement of the right leg SFI (injured side) was coded "experimental" (E) while the left leg (uninjured side) was coded "normal" (N).

The SFI was calculated using the following formula: SFI = -38.3 (EPL—NPL)/NPL + 109.5 (ETS—NTS)/ NTS + 13.3 (EIT—NIT)/NIT—8.8. Two examiners blinded to the experimental procedure performed the SFI evaluations. An SFI value which near to -100 indicates complete dysfunction, whereas a value near to 0 indicates normal motor function.

## Analysis of the target muscle weight (gastrocnemius muscle)

After 12 weeks of evaluation, the experimental rats were sacrificed. The gastrocnemius muscles of the normal and experimental sides were dissected, then the weight of the gastrocnemius muscle was measured using Camry electronic pocket scale model EHA-401. The wet weight ratio of the gastrocnemius muscle was calculated by dividing the weight of the muscle on the experimental side by the normal side.

## Histomorphometry examination and sciatic nerve defect regeneration

The histomorphometry assay is a quantitative method to measure the size and the amount of a structure in a tissue by using a microscope which is equipped with computerized image analysis. Twelve weeks after the surgery, the defect of nerve segment which has been regenerated was harvested and fixated in 4% paraformaldehyde solution. Microscopic preparation was made out of the segment. The sciatic nerve was placed on an object glass covered with silane (BIOGEAR Microscope Slide). It was labeled with code so that the examiner was blinded of the specimen that they analyzed. The histological staining of toluidine blue (TB) were used for the preparation of sciatic nerve specimen to assess the number and diameter of the myelinated axon.

Myelinated axons were counted quantitatively by using light microscope Leica type CXDM750 AMD-7536. All visual fields of one axial cut of the sciatic nerve that taken from the mid portion of the regenerated sciatic nerve were measured. They were merged into one complete image (photomerge) by using PTGui Pro 9.1. The amount and diameter of myelinated axons was measured using image-J program (Image J is a public program developed by Research Service Branch (RSB) of National Institute of Mental Health (NIMH) for image processing).

## Statistical analysis

The functional and histomorphometric analyses were performed by observers blinded to each group. Data were expressed as mean–standard deviation (SD). The statistical differences of SFI results, the weight of the left gastrocnemius muscle, the weight of the right gastrocnemius muscle, the ratio of gastrocnemius muscle weight, the number of myelinated axons, and the diameter of myelinated axon among all groups of treatment were evaluated using one way analysis of variance (ANOVA). Post hoc Bonferroni test was performed to discover the statistically differences between each group. The p value of $<0.05$ was considered statistically significant.

## Results

### UC-CM enhances functional recovery (sciatic functional index/SFI)

The functional recovery at nine week after procedure (SFI 2) was better in group I (-49.7 ± 1.8) and group III (-48.2 ± 4.2) than the other groups which mean the recovery of sciatic nerve

**Table 1.** SFI results in each treatment group.

| Parameter | Treatment Groups | | | | p value* |
|---|---|---|---|---|---|
| | *Autograft (group I)* | *Umbilical Cord MSC (group II)* | *Secretome (group III)* | *Conduit (group IV)* | |
| SFI 1 | -78.2 ± 3.6 | -60.4 ± 4.7 | -56.8 ± 6.7 | -85.3 ± 1.0 | 0.001 |
| SFI 2 | -49.7 ± 1.8 | -62.6 ± 2.6 | -48.2 ± 4.2 | -73.0 ± 1.9 | <0.001 |
| SFI 3 | -32.8 ± 4.5 | -40.3 ± 1.3 | -33.5 ± 2.2 | -47.7 ± 2.8 | 0.008 |

function was faster in those groups compared to others (Table 1). The SFI 3 was higher in group I and III compared to other groups (p = 0.008). Group IV showed the slowest functional recovery when compared to the other treatment groups at sixth week (p = 0.001) and ninth week of evaluation (p≤0.001). This group also showed the worst motoric function recovery at the end of the research (p = 0.008).

Moreover post hoc Bonferroni test showed significantly higher motoric function recovery in group III compared to group I (p = 0.02) while there is no significant difference in group II and IV compared to group I at sixth week. The next evaluation in ninth week found that there was no significant difference between group III and group I which indicated the recovery of motoric function after secretome implantation resembling the gold standard autograft. Moreover the SFI 2 results of Group III was significantly better than Group II (p = 0.01).

## UC-CM improves the ratio of target muscle weight and prevents muscle atrophy

After sciatic nerve transection, the gastrocnemius muscles atrophied and subsequently the gastrocnemius muscle reinnervated by sciatic nerve regeneration. The gross morphology of gastrocnemius muscle in group I and III were bigger than group II and IV (Fig 2). The muscles exhibited slim gross morphology in group IV.

The greatest weight of the right gastrocnemius muscle (injured side) were found in group I (1.8 ± 0.3 g) and it is not significantly different with group III (1.4 ± 0.5 g), showed in Fig 3. This finding suggested that the reinnervation of sciatic nerve to the target muscle in secretome group was similar when compared to autograft group. The right gastrocnemius muscle weight in secretome group was greater than UC-MSC group (p = 0.002). Nonetheless the weight of left gastrocnemius muscle in four groups were not significantly different which showed the similar weight of normal gastrocnemius muscle in all rats.

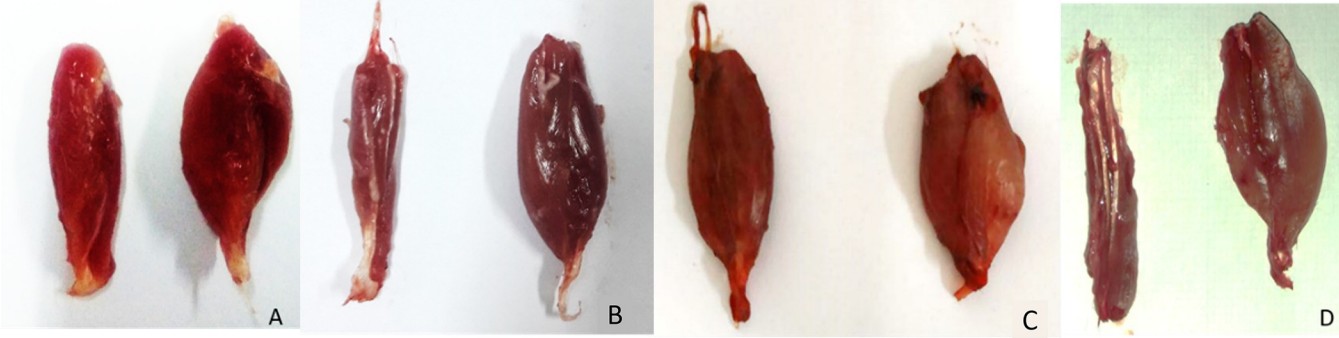

**Fig 2.** Gross morphology of gastrocnemius muscle from both legs in four treatment groups (A. autograft, B. secretome, C. UC-MSC, D. conduit only). Gastrocnemius on left side of figure was the injured side (sciatic nerve defect) and right side of figure was the normal side.

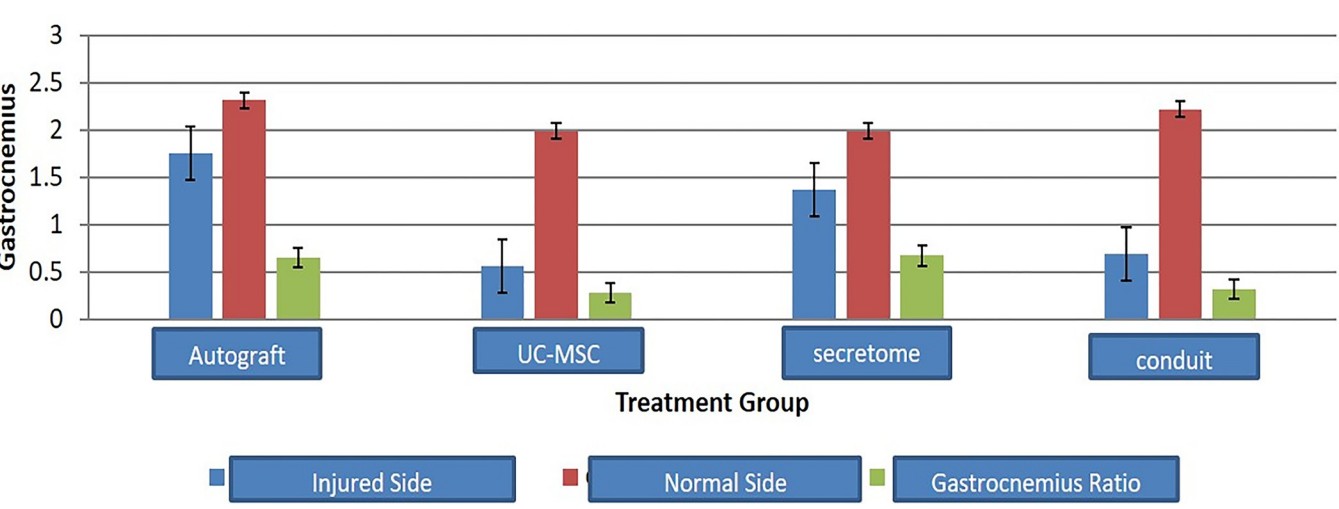

**Fig 3. Gastrocnemius muscle ratio in each treatment group.**

The ratio of gastrocnemius muscle weight was higher in the reinnervated muscle of the autograft and secretome group, namely 0.7 ± 0.1 and 0.7 ± 0.2 when compared with that of the UC-MSC and conduit only group, namely 0.3 ± 0.1 and 0.3 ± 0.2 (p<0.001, Fig 3). Moreover, the ratio of the gastrocnemius muscle weight in secretome group was higher than UC-MSC group (p<0.001).

## UC-CM stimulates peripheral nerve regeneration

There was no dislocation of the autograft or conduit found at 12 weeks after surgery. However, not all sciatic nerve defect of SD rats in four groups was regenerated which were one sciatic nerve defect in each of UC-MSC and conduit only group. Neither of these non regenerated nerves formed neuromas. All of SD rats in group I demonstrated well-incorporation of autograft. Gross morfology of regenerated nerves showed thicker sciatic nerve in secretome group than those in UC-MSC group and conduit only group (Fig 4). The conduit only group demonstrated the less growth of sciatic nerve.

The mid portion of regenerated sciatic nerve was harvested and stained using toluidine blue for histomorphometry examination. The examination revealed significant difference in the number of regenerated myelinated axon in each group (p<0.001; Fig 5). The largest number of myelinated axons was found in group I (175.1 ± 15.9), followed by group III (168.9 ± 11.7); group II (134.7 ± 3.6); and group IV (126.0 ± 0.3). Post hoc Bonferroni test revealed no significant difference in the number of myelinated axons between group I and group III. However, the number of myelinated axons in group III was significantly higher than group II (p = 0.002).

In addition, Pearson correlation test revealed a significant high correlation between the number of myelinated axons and the weight of right gastrocnemius muscle (r = 0.9); the ratio of gastrocnemius muscle (r = 0.8); SFI 2 (r = 0.6). There was also a significant moderate correlation between the number of myelinated axons and SFI 3 (r = 0.5). On the other hand, no correlation was found between the number of myelinated axons and SFI 1.

Eventhough the secretome enhanced the number of regenerated myelinated axons, it seemed that secretome had no influence on the diameter of regenerated myelinated axons.

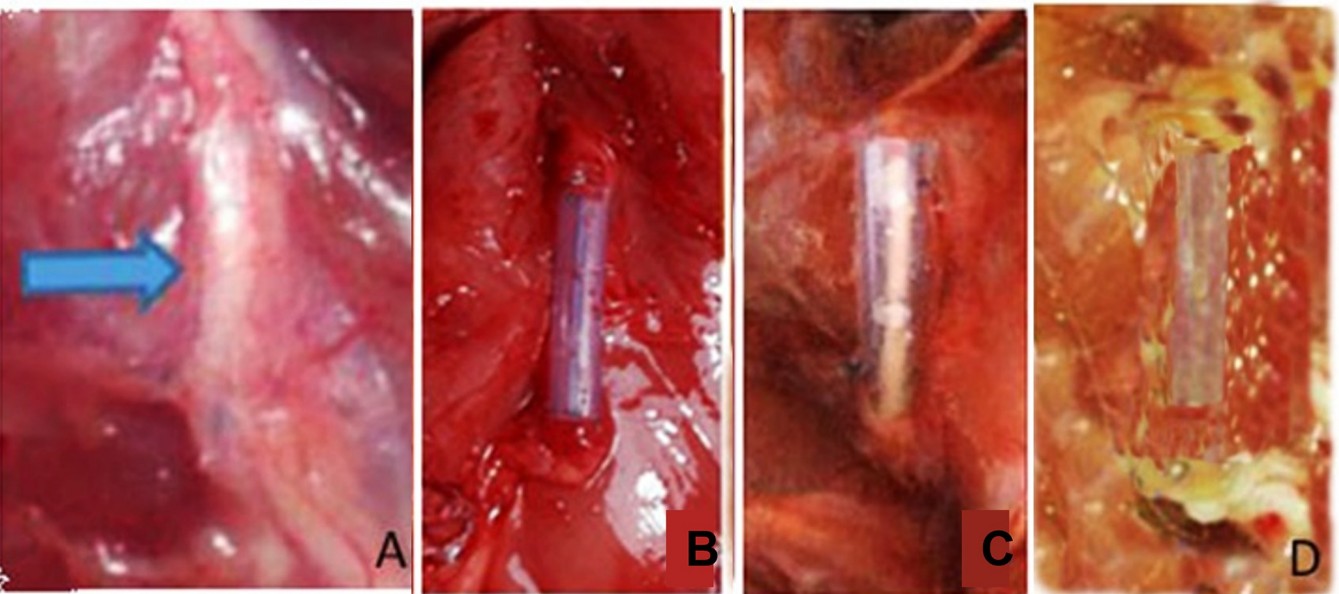

**Fig 4.** Harvested sciatic nerve in (A) autograft, (B) secretome, (C) UC-MSC, and (D) conduit only group.

The histomorphometry examination showed no significant difference in diameter of regenerated myelinated axons in each group. The largest diameter of myelinated axon was found in group I (5.0 ± 0.2 μm), followed by group III (4.6 ± 0.4 μm); group II (4.1 ± 0.6 μm); and group IV (4.4 ± 1.1 μm).

## Discussion

The findings in this study show that all treatment groups exhibited sciatic nerve regeneration, but the speed and rate of motor function recovery in each treatment group differed from the other groups. The ultimate goal of peripheral nerve defect treatment is the functional recovery [3]. However, it is not easy to evaluate motor funtion recovery in SD rats model. Therefore, indirect methods was used such as the SFI examination and the gastrocnemius muscle wet weight ratio [9,10].

The SFI 1 result in secretome group was better than autograft group (Table 1). This phenomenon may caused by the presence of growth factors, chemokins, and cytokines in the

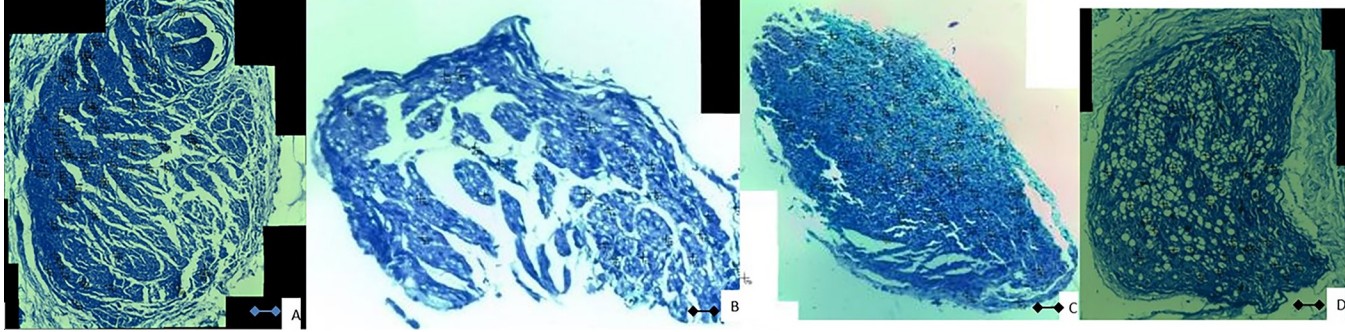

**Fig 5. Histomorphometry examination of the axial cut of regenerated myelinated axon at 12 weeks after surgery. Scale bar = 30 μm.** (A) autograft, (B) secretome, (C) UC-MSC, and (D) conduit only.

secretome, which are necessary for regeneration of the sciatic nerve in early phase and modulation of inflammatory process so that generates an optimal milieu for peripheral nerve regeneration thus allow the healing to be faster. Sugimura-wakayama et al. reported that secretome may contained BDNF (brain-derived neurotrophic factor), NGF (nerve growth factor), GDNF (glial cell line-derived neurotrophic factor), CNTF (ciliary neurotrophic factor), NT-3 (neurotrophin-3), HGF (hepatocyte growth factor), and VEGF (vascular endothelial growth factor) [3,6–8]. Other studies confirmed that the secretome of umbilical cord contained placental growth factor (PlGF), EGF (endothelial growth factor), NGF, dan VEGF in significant amount that had an advantageous microenvironment to facilitate nerve regeneration [11,12]. Moreover the application of multiple growth factors rather than single factors showed promising therapeutic effect [13–15].

At six weeks after surgery, lower SFI 1 values in other groups could be understood because the process of sciatic nerve regeneration was still ongoing and there were still parts of nerve defect that had not undergone regeneration yet. At nine weeks after surgery, tremendous improvement of SFI 2 in autograft group took place and only secretome group could keep up with them (Table 1). This finding exhibited that the secretome effect in enhancing peripheral nerve regeneration was similar to the gold standard treatment of the peripheral nerve defect. Moreover the SFI 2 of secretome group showed significantly better result than UC-MSC group which may explained due to drastic reduction of remaining viable MSC in the milieu of affected nerve [16–18]. Furthermore, secretome may also improved functional recovery by increasing sciatic nerve conduction rate recovery through enhancement of axon myelination [19,20].

At twelve weeks after surgery, conduit only group generated a significantly poor functional outcome compared to other groups. This probably was caused by poor myelination of the axons and degeneration of end plate had been occured before the regeneration of axons took place. Myelination is an essential step in nerve healing process since it allows the electrical impulse propagated along the axon to be transmitted faster, since the myelin sheath maintains the electrical impulse within the axonal membrane [19,20]. The periodically SFI evaluation in this research was adjusted to the peripheral nerve recovery process in SD rats that required 12–20 weeks after the nerve injury [16,21,22].

Autograft and secretome group had greater gastrocnemius muscle weight ratio compared to the other groups [3,16]. This suggested that the best reinnervation process of sciatic nerve took place in autograft and secretome group, so that prevented the gastrocnemius muscle atrophied in those groups. Beside allowing faster recovery when compared to UC-MSC and conduit only, secretome was also proven to reduce the target muscle atrophy which also suggested enhancement of muscle function and consequently better motor functional recovery [16–18].

Histomorphometry examination of the regenerated sciatic nerve segment revealed that the greatest number of myelinated axons was in autograft and secretome groups. This finding is in accordance with the clinical parameters (gastrocnemius muscle ratio and SFI). The number of myelinated axons in secretome group was greater than UC-MSC and conduit only group. Other studies showed low MSC viability in the neural milieu,mostly the cells only lasted for one to two weeks [3,4]. Only a small percentage of the MSC differentiated and integrated effectively with the host cells [23–25].

Besides the secretome enhanced the peripheral nerve regeneration relative comparably to the autograft, the use of secretome had several advantages compared to MSC including [26,27]:

a. The application of secretome can overcome several safety problems that can potentially be related to transplantation of living and proliferating cell populations including immune

compatibility, possible tumor formation, embolism formation, and transmission of infection;

b. Secretome can be evaluated for safety, dosage, and potential benefits which are basically analogous to conventional pharmaceutical agents; (3) secretome is more economical and practical for clinical applications where it does not require invasive cell collection procedures;

c. The storage process can be carried out without the use of potentially toxic cryopreservation agents in order to maintain potential product;

d. Large quantities of production are feasible under controlled laboratory conditions;

e. The secretome used in this study is MSC culture waste which is usually disposed of. So, in addition to the MSC, the waste can also be utilized, whose production does not require additional cost;

f. Secretome stock can be available for the treatment of acute conditions such as trauma cases while it is not feasible in MSC preparation;

g. Secretome can provide more convenience compared to MSC in terms of manufacturing, storage, handling, shelf life, and the potential as a biological therapeutic agent.

## Conclusion

The secretome showed comparable results to autografts in regenerating sciatic nerve defects, with faster regeneration preventing end plate degeneration and target muscle atrophy. It outperformed umbilical cord MSC and conduit-only groups in peripheral nerve regeneration. Secretome implantation is quicker and overcomes challenges associated with autografts, such as donor-related issues and procedural complexity. Formulated secretome has the potential to surpass autografts in peripheral nerve defect regeneration, making it a novel therapy to replace autografts in clinical management.

## Author Contributions

**Conceptualization:** Aryadi Kurniawan, Ismail Hadisoebroto Dilogo, Jeanne Adiwinata Pawitan, Wahyu Widodo, Ade Martinus.

**Data curation:** Aryadi Kurniawan, Jeanne Adiwinata Pawitan, Wahyu Widodo, Ade Martinus.

**Formal analysis:** Aryadi Kurniawan, Jeanne Adiwinata Pawitan, Wahyu Widodo, Ade Martinus.

**Investigation:** Ismail Hadisoebroto Dilogo, Jeanne Adiwinata Pawitan, Ihsan Oesman.

**Methodology:** Aryadi Kurniawan, Jeanne Adiwinata Pawitan.

**Project administration:** Wahyu Widodo, Ihsan Oesman.

**Resources:** Aryadi Kurniawan, Jeanne Adiwinata Pawitan, Ihsan Oesman.

**Supervision:** Ismail Hadisoebroto Dilogo, Jeanne Adiwinata Pawitan, Wahyu Widodo, Ihsan Oesman.

**Validation:** Aryadi Kurniawan, Ismail Hadisoebroto Dilogo, Jeanne Adiwinata Pawitan.

**Writing – original draft:** Ade Martinus.

**Writing – review & editing:** Aryadi Kurniawan, Ismail Hadisoebroto Dilogo, Wahyu Widodo, Ihsan Oesman, Ade Martinus.

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
