## [Decision Letter · Decision Letter 0]

6 May 2024

PONE-D-24-15970The Effects of Secretome of Umbilical Cord Mesenchymal Stem Cells on Regeneration of Sciatic Nerve Defects in Sprague Dawley RatPLOS ONE

Dear Dr. Martinus,

Thank you for submitting your manuscript to PLOS ONE. After careful consideration, we feel that it has merit but does not fully meet PLOS ONE’s publication criteria as it currently stands. Therefore, we invite you to submit a revised version of the manuscript that addresses the points raised during the review process.

We look forward to receiving your revised manuscript.

Kind regards,

Yung-Hsiang Chen, Ph.D.

Academic Editor

PLOS ONE

Journal Requirements:

The name of the colleague or the details of the professional service that edited your manuscript.A copy of your manuscript showing your changes by either highlighting them or using track changes (uploaded as a *supporting information* file).A clean copy of the edited manuscript (uploaded as the new *manuscript* file).

4. We note that your Data Availability Statement is currently as follows: 

"All relevant data are within the manuscript and its Supporting Information files"

7. Please include a copy of Table 1 which you refer to in your text on pages 6, 8, 9 ad 14.

**Additional Editor Comments:**

Thank you for submitting the following manuscript to PLOS ONE.

Please revise the manuscript according to the reviewers' comments and upload the revised file.

Reviewers' comments:

Reviewer's Responses to Questions

**Comments to the Author**

1. Is the manuscript technically sound, and do the data support the conclusions?

Reviewer #1: No

Reviewer #2: Yes

2. Has the statistical analysis been performed appropriately and rigorously? 

Reviewer #1: Yes

Reviewer #2: Yes

3. Have the authors made all data underlying the findings in their manuscript fully available?

Reviewer #1: No

Reviewer #2: Yes

4. Is the manuscript presented in an intelligible fashion and written in standard English?

Reviewer #1: Yes

Reviewer #2: Yes

5. Review Comments to the Author

Reviewer #1: This is a report of effect of secretome for peripheral nerve regeneration. However, I cannot consider this paper is suitable for publication. There are many issues to be addressed by the authors.

Line 93, the author should evaluate the content of CM.

Line 109-111, the author should add the references of SFI evaluation.

Line 122, the authors should add the explanation how they prepare the histological sections.

Line 128-129, the author should add the explanation to measure the axon number and diameter, and also add the references of the methodology.

Line 144, the author should add Table 1. A few values are on the manuscript. I cannot judge the statistical results.

Line 157, the order of the pictures in Figure 2 is different from the group oreder. The author should arrange the picture order according to group number.

Line 165-166, the value of the ration is rough. The author should recalculate the ratio down to third decimal.

Line 177, the pictures in Figure 5 are rough. The author should add the picture in high magnification. They should also add the electron microscopical picture.

In Materials and Methods, the author should add the experiment how secretom works. SFI, muscle weight and morphological study are not enough to explain UC-CM stimulates peripheral nerve regeneration.

Figure 1, there is no explanation for the fourth picture.

Figure 2, the author should arrange the picture order according to group number.

Figure 4, the author should arrange the picture order according to group number. The regenerated nerve cannot be recognized in B, C or D.

Figure 5, the pictures are rough. It is impossible to count the axons in those pictures. The author should add the picture in high magnification. They should also add the electron microscopical picture.

Reviewer #2: The discussion could be enriched using the following reference:

The Long-term Effects of Uncultured Omental Adipose-derived Nucleated Cells Fraction and Bone-marrow Stromal Cells on Sciatic Nerve Regeneration

Saeed Azizi; Behnam Heshmatian; Saman Mahmoudpour; Abbas Raisi

Volume 10, Issue 2, December 2015, Pages 21-29

6. PLOS authors have the option to publish the peer review history of their article (what does this mean?). If published, this will include your full peer review and any attached files.

Reviewer #1: No

Reviewer #2: **Yes: **Rahim Mohammadi

---

## [Author Response · Author response to Decision Letter 0]

14 Aug 2024

Dear reviewers, we have revised the manuscript according to your comments. The detailed responses are attached in the "Response to Reviewers" file.

---

## [Decision Letter · Decision Letter 1]

2 Sep 2024

The Effects of Secretome of Umbilical Cord Mesenchymal Stem Cells on Regeneration of Sciatic Nerve Defects in Sprague Dawley Rat

PONE-D-24-15970R1

Dear Dr. Martinus,

We’re pleased to inform you that your manuscript has been judged scientifically suitable for publication and will be formally accepted for publication once it meets all outstanding technical requirements.

Kind regards,

Yung-Hsiang Chen, Ph.D.

Academic Editor

PLOS ONE

Additional Editor Comments (optional):

Congratulations on the acceptance of your manuscript, and thank you for your interest in submitting your work to PLOS ONE.

Reviewers' comments:

Reviewer's Responses to Questions

**Comments to the Author**

1. If the authors have adequately addressed your comments raised in a previous round of review and you feel that this manuscript is now acceptable for publication, you may indicate that here to bypass the “Comments to the Author” section, enter your conflict of interest statement in the “Confidential to Editor” section, and submit your "Accept" recommendation.

Reviewer #2: All comments have been addressed

Reviewer #3: All comments have been addressed

2. Is the manuscript technically sound, and do the data support the conclusions?

Reviewer #2: Yes

Reviewer #3: Yes

3. Has the statistical analysis been performed appropriately and rigorously? 

Reviewer #2: Yes

Reviewer #3: Yes

4. Have the authors made all data underlying the findings in their manuscript fully available?

Reviewer #2: Yes

Reviewer #3: Yes

5. Is the manuscript presented in an intelligible fashion and written in standard English?

Reviewer #2: Yes

Reviewer #3: Yes

6. Review Comments to the Author

Reviewer #2: Dear Authors,

I hope this message finds you well.

I am writing to inform you that I have reviewed the revised version of your manuscript titled "Experimental study of cBMMSC based on nanosilver hydrogel nerve conduit for repairing spinal cord injury." I would like to commend you on the thorough revisions and the improvements made to the manuscript. The changes have significantly enhanced the clarity and quality of your work.

Based on my review, I am pleased to inform you that the revised manuscript meets the standards required for publication. I believe it makes a valuable contribution to the field, and I am confident it will be of interest to the journal's readers.

Thank you for your hard work and responsiveness to the feedback provided. I wish you the best with the final stages of the publication process.

Reviewer #3: Current treatments for peripheral nerve defects are less than ideal. This study examines the regenerative potential of the secretome from umbilical cord mesenchymal stem cells (UC-MSCs) for addressing peripheral nerve defects. The research involved 28 Sprague Dawley rats with sciatic nerve injuries, which were divided into four groups, each receiving a different treatment: autografts, MSC implantation, UC-MSC secretome, or silicone conduits. The results showed that the UC-MSC secretome was similarly effective as autografts in promoting nerve regeneration, with significant improvements in muscle weight ratio and the number of myelinated axons. UC-MSC secretome could serve as a novel therapy for peripheral nerve defects, potentially replacing autografts. This paper has already undergone review, and the revisions were effectively executed.

7. PLOS authors have the option to publish the peer review history of their article (what does this mean?). If published, this will include your full peer review and any attached files.

Reviewer #2: **Yes: **Rahim Mohammadi

Reviewer #3: No

---

## [Editor Report · Acceptance letter]

23 Oct 2024

PONE-D-24-15970R1 

PLOS ONE

Dear Dr. Martinus, 

I'm pleased to inform you that your manuscript has been deemed suitable for publication in PLOS ONE. Congratulations! Your manuscript is now being handed over to our production team.

Kind regards, 

on behalf of

Dr. Yung-Hsiang Chen 

Academic Editor

PLOS ONE